# The Adenosinergic Pathway in Non-Small Cell Lung Cancer

**DOI:** 10.3390/cancers16183142

**Published:** 2024-09-13

**Authors:** Olivier Van Kerkhove, Saartje Verfaillie, Brigitte Maes, Kristof Cuppens

**Affiliations:** 1Department of Pulmonology and Thoracic Oncology and Jessa & Science, Jessa Hospital, Salvatorstraat, 3500 Hasselt, Belgium; 2Laboratory for Molecular Diagnostics, Department of Laboratory Medicine, Jessa Hospital, Salvatorstraat, 3500 Hasselt, Belgium; 3Faculty of Medicine and Life Sciences-LCRC, Hasselt University, 3590 Diepenbeek, Belgium

**Keywords:** lung cancer, immune checkpoint inhibition, adenosine

## Abstract

**Simple Summary:**

Lung cancer remains the most important cause of cancer-related mortality worldwide. Immune checkpoint inhibitors revolutionized lung cancer care. These molecules restore the host’s immune response against tumor cells and led to impressive results in non-small cell lung cancer patients. However, these benefits are only observed in a minority of patients. Extracellular adenosine is an immune checkpoint that contributes to immune evasion in tumor cells. Current research focuses on targeting this pathway with the aim of inducing durable treatment effects in a greater proportion of lung cancer patients.

**Abstract:**

Immune checkpoint inhibitors (ICIs) targeting PD-(L)1 and CTLA-4 have revolutionized the systemic treatment of non-small cell lung cancer (NSCLC), achieving impressive results. However, long-term clinical benefits are only seen in a minority of patients. Extensive research is being conducted on novel potential immune checkpoints and the mechanisms underlying ICI resistance. The tumor microenvironment (TME) plays a critical role in modulating the immune response and influencing the efficacy of ICIs. The adenosinergic pathway and extracellular adenosine (eADO) are potential targets to improve the response to ICIs in NSCLC patients. First, this review delves into the adenosinergic pathway and the impact of adenosine within the TME. Second, we provide an overview of relevant preclinical and clinical data on molecules targeting this pathway, particularly focusing on NSCLC.

## 1. Introduction

Lung cancer is the leading cause of cancer-related death worldwide [1]. Treatment of patients with non-small cell lung cancer (NSCLC) has improved significantly during the last decade after the introduction of immune checkpoint inhibitors (ICIs) [2,3,4]. Inhibitors of programmed death protein 1 (PD-1), its ligand (PD-L1), and, to a lesser extent, cytotoxic lymphocyte antigen 4 (CTLA-4) are now part of the standard of care in lung cancer. Several practice-changing trials showed an impressive improvement in clinical outcomes in patients treated with ICIs (alone or in combination with chemotherapy) compared to chemotherapy alone [2,3,4,5,6,7,8,9]. Furthermore, long-term durable responses are seen in a subset of patients by eliciting a potent antitumoral immune response. Despite these impressive results in a sizable minority of patients, the majority either do not respond to ICIs at all or develop resistance during treatment. The landmark KEYNOTE-024 trial, in which pembrolizumab (inhibitor of PD-1) was compared to chemotherapy in patients with high PD-L1 expression, showed an objective response rate (ORR) of only 44.8% in the patient group treated with ICIs. In addition, more than 50% of the patients who received pembrolizumab developed disease progression at 12 months [2,3,4,5,6,7,8,9].

The tumor microenvironment (TME) is crucial in cancer immunity and has become the prime subject of several studies focusing on primary and acquired resistance to ICIs [10,11]. Tumor cells, immune cells, vessels, signaling mediators, and stromal cells are all part of the TME [12,13]. In particular, the role of antigen-presenting cells (APCs) and tumor-infiltrating lymphocytes (TILs) has been extensively studied. The presence of TILs positively correlates with immunogenicity and hence response to therapy and survival [14]. However, the mere presence of TILs is not a guarantee for improved outcome to therapy. Differentiation of TILs is equally important: CD8^+^ T cells correlate with a better prognosis, while regulatory T cells (Tregs), which have immunosuppressive features, tend to lead to a worse prognosis. Macrophages in the TME can harbor proimmunogenic as well as immunosuppressive features. The M2 phenotype promotes tumor growth and invasiveness and is associated with a worse prognosis. The M1 phenotypic counterpart has a more favorable outcome in NSCLC. Myeloid-derived suppressor cells (MDSC) also exhibit immunosuppressive function and correlate with an unfavorable prognosis in NSCLC [15]. Another relevant structure within the TME is tertiary lymphoid structures (TLSs), which are composed of T cell and B cell zones. A larger area of TLSs and an increased B cell proportion seem to correlate with a better survival in cancer patients [13,16,17].

Extracellular adenosine (eADO) can suppress the activity of CD8^+^ T cells, as discovered in 1975 [18]. Multiple studies after that confirmed that eADO can suppress antitumor immunity [19,20]. The adenosinergic pathway, which contributes to immune evasion, is often exploited in lung cancer, particularly in oncogenic-driven lung adenocarcinoma (LUAD). Oncogene-driven LUAD, such as lung cancer harboring activating epidermal growth factor receptor (EGFR) mutations, is known to be fairly resistant to ICI therapy, which can be at least partially explained by the adenosine pathway. The extent to which eADO promotes immune evasion in other lung cancer types, such as non-oncogene driven LUAD, squamous carcinoma (LUSC), and small-cell lung carcinoma (SCLC), is unclear [21,22,23].

Alterations in the adenosine pathway impact the prognosis of NSCLC as well. Higher CD73 (one of the enzymes responsible for eADO production) levels appear to correlate with poor prognosis. Paradoxically, high expression of the A2_A_ receptor (one of the functional adenosine receptors) on tumor cells is associated with a more favorable prognosis. These findings are somewhat discrepant, and a clear explanation is currently lacking [24].

Because of these findings and the peculiarities of the adenosine pathway, which we will further elaborate in this manuscript, eADO and, by extension, the entire adenosinergic pathway, are potential targets for the development of new anti-lung cancer drugs. We will review the biology of eADO and its role in the TME of NSCLC, discuss relevant preclinical data, and highlight early clinical data on adenosine and its pathway.

## 2. Generation and Metabolism of Adenosine

### 2.1. Adenosine Is Formed by a Canonical and a Non-Canonical Pathway

Two main pathways are responsible for eADO generation (Figure 1). In the canonical pathway, eADO is formed through hydrolysis of adenosine triphosphate (ATP) [20,25]. In contrast to adenosine, extracellular ATP has a proinflammatory role in the TME, where it is released in response to hypoxia, ischemia, and inflammation [26]. CD39 (ectonucleoside triphosphate diphosphydrolase 1) is responsible for the conversion of ATP to adenosine diphosphate (ADP) and eventually adenosine monophosphate (AMP) [27]. Extracellular AMP (eAMP) is then converted to eADO by CD73 (5′-nucleotidase) [25]. An alternative method of eADO generation is the non-canonical pathway, in which nicotinamide adenine dinucleotide (NAD+) is used as a precursor to generate eAMP by CD38 (NAD+ glycohydrolase) and CD203 (ectonucleotide pyrophosphatase/phosphodiesterase 1). eAMP is then, using the common canonical pathway, converted to eADO by CD73 [25,28].

### 2.2. Hypoxia Induces the Formation of eADO

Earlier studies showed that hypoxia plays a pivotal role in the upstream part of the adenosinergic pathway [29]. When oxygen is abundant, hypoxia-inducible factor (HIF)-1α and HIF-2α can bind the Von Hippel-Lindau (VHL) gene product, leading to its proteasomal degradation. This binding is mediated by hydroxylation of proline residues on HIF-α [29,30]. Hydroxylation is less efficient under hypoxic conditions, as seen in inflammatory environments or in the TME, which leads to the stabilization of HIF-α. HIF-α then forms a complex with HIF-1β in the nucleus, and this complex binds the hypoxia-responsive elements (HREs) of the promotor region of specific genes needed for adaptation under hypoxic circumstances [31,32,33]. The promotor region of the genes that encode for CD73 and two central adenosine receptors (A2_A_ and A2_B_) feature such HREs [34,35,36]. This illustrates the primary link between hypoxia and adenosine signaling.

### 2.3. The Formation of eADO Is Upregulated within the TME

The concentration of adenosine in the TME lies within micromolar ranges [20]. ATP, the precursor of eADO in the canonical pathway, is found in high concentrations in the TME. This is due to passive release after cell death on one hand, and active secretion by the tumor on the other hand, in response to hypoxia and inflammation [26]. ATP, unlike eADO, is proinflammatory and can promote antitumor immune responses [26].

Conversion of ATP to eADO, is facilitated by the ectonucleotidases CD39 and CD73, which are typically upregulated in cancer cells. CD73, as a promotor of immune suppression, has been studied in oncogene-driven NSCLC. Han et al. demonstrated a significant upregulation of CD73 in cells harboring an EGFR mutation, KRAS mutation, or ALK re-arrangement, as frequently seen in NSCLC. The ERK-Jun and c-Jun pathways were mainly involved in this upregulation. They also showed that selective inhibition of these driver proteins led to deregulation of CD73. Another preclinical study showed an upregulation of the CD73 gene panel in EGFR-mutated NSCLC using single-cell analysis [21,22]. A2_A_ and A2_B_ receptor expression is also increased in cancer. The underlying mechanism is roughly the same as that of CD39 and CD73. Hypoxia and transforming growth factor β (TGFβ) play a crucial role in this induction [37]. TGFβ also stimulates the expression of CD39 and especially CD73 on the surface of T cells, dendritic cells (DCs), macrophages, and MDSCs, as shown by several studies [38,39].

Another condition accompanied by high levels of eADO is a specific form of tumor dedifferentiation called epithelial-to-mesenchymal transition (EMT). EMT enables tumor cells to migrate and invade other structures, thus leading to metastasis. During EMT, epithelial cells lose some epithelial functions (e.g., tight junctions and apicobasal polarity) and gain mesenchymal features, making the cells more motile and invasive. EMT can be induced by multiple factors, including the earlier mentioned TGFβ pathway and activation of the WNT/β-catenin pathway [40,41]. The WNT/β-catenin pathway is able, just like TGFβ, to induce expression of CD73, leading to higher concentrations of eADO in the TME [38,39,42]. Lupia et al. showed in ovarian cancer cell lines that CD73 can promote the expression of EMT-associated genes as well, thus creating a feedforward loop [43].

There are also indications that the non-canonical pathway based on CD38 may be upregulated in the TME. Higher concentrations of CD38-mediated eADO were seen in more aggressive human multiple myeloma cell lines [44]. Furthermore, CD38-mediated immunosuppression is a possible mechanism of tumor escape when treated with PD-(L)1 blockade [45].

### 2.4. eADO Is Either Degraded in the Extracellular Space or Transported Intracellularly

Extracellular adenosine is cleared out of the extracellular space by multiple mechanisms. eADO, at excessive concentrations, can be degraded to inosine by adenosine-deaminase (ecto-ADA), a membrane-associated enzyme [46]. eADO can also be transported intracellularly through both equilibrative and concentrative transporters. Different enzymes are responsible for the intracellular metabolism of adenosine, such as adenylate kinase (ADK). The latter can be inhibited by HIF, leading to higher concentrations of eADO in the case of hypoxia and inflammation, as seen in the TME.

### 2.5. Overview of the Four Adenosine Receptors

eADO functions through four distinct G protein-coupled receptors: A1, A2_A_, A2_B_, and A3. The human A1, A2_A_, and A3 receptors are all high-affinity receptors, whereas the A2_B_ receptor is a low-affinity receptor and is thus only activated in pathological conditions with elevated eADO concentration. Signal transduction of these receptors occurs via cAMP, which is inhibited by the A1 and A3 subtypes and stimulated by both A2 subtypes [47]. Immunosuppression in the TME is mainly exerted by the A2_A_ and A2_B_ receptors; hence, our review will focus on these two receptor subtypes [48,49].

## 3. Functions of eADO

### 3.1. eADO and Immune Cells

Extracellular adenosine can induce immunosuppression through a variety of immune cells (Figure 2). First, eADO leads to impairment of CD8^+^ lymphocytes. The mTOR (mammalian target of rapamycin) pathway plays a key role in the differentiation and activation of T cells [50]. Signaling through the A2_A_ receptor (A2AR) activates protein kinase A (PKA), leading to reduced activation of the mTORC1 pathway, impairing immunocompetence by, among other mechanisms, hampering T-cell receptor (TCR) functions. The A2AR/PKA/mTOR2C pathway is presumably the main signal route by which eADO exerts immunosuppressive effects on CD8+ T cells [51].

eADO also inhibits effector T-cell function by upregulation of several immune checkpoints, such as PD-1, lymphocyte activation gene 3 protein (LAG3), and T cell and immunoglobulin and mucin domain-containing protein 3 (TIM3) [52].

As previously mentioned, CD8^+^ T cells are reduced in the TME of EGFRm LUAD. Inhibition of CD73 restores T cell presence in the TME [22]. MET amplification often co-occurs with EGFRm NSCLC and is a known mechanism of acquired resistance to EGFR TKIs. MET amplification in EGFRm LUAD activates the stimulator of interferon genes (STING), an emergent determinant of innate cancer immunogenicity [53]. However, CD73 is significantly upregulated, leading to adenosinergic immunosuppression constraining STING. The combination of CD73 inhibition and pemetrexed, a known potentiator of STING, has been shown to enhance CD8^+^ T cell immunity in humanized mouse models [54].

The adenosine pathway affects various other immune cells. Signaling through the A2_A_ receptor promotes the differentiation of CD4^+^ T cells to FoxP3^+^ Treg cells, which have a known immunosuppressive function [55]. eADO impairs the maturation of NK cells, and preclinical data have shown restoration of NK cell function and improved tumor control in A2_A_ receptor-deficient mice [56]. cAMP is able to suppress the transcription of NF-κβ, downstream of both the B cell receptor and Toll-like receptor 4. This leads to an impaired activation and survival of B lymphocytes [57]. In macrophages, adenosine signaling promotes differentiation to the tolerogenic M2-phenotype, characterized by VEGF and IL-10 expression, which in turn promotes tumor growth [58]. The adenosine pathway also leads to more tolerogenic DCs that are far less capable of priming CD8^+^ T cells [59]. Finally, MDSCs are a subset of immature myeloid cells with a regulatory function. These cells contribute to the immunosuppressive features of the TME. eADO stimulates the expansion of MDSCs via its A2_B_ receptor [60,61].

### 3.2. eADO and Tumor Cells

eADO promotes tumor growth and metastasis in LUAD [62]. A specific link between adenosine and NSCLC-related bone metastases has been described. Both CD73 and A2 receptors were upregulated in NSCLC stem cells, which preferentially metastasize to the bone [63]. The same involvement of eADO in the process of (bone) metastasis was also shown in other tumor types [64,65].

CD73 and eADO can stimulate mesenchymal transition, as mentioned before, and can create a feedforward loop [43]. CD73 also has other protumor effects independent of its enzymatic function. It was shown that CD73 directly binds to extracellular matrix proteins, such as tenascin C, to enhance the adhesiveness and invasiveness of melanoma cells [66].

Tumor growth, infiltration, and metastasis are therefore stimulated by the adenosinergic pathway.

### 3.3. eADO and Other Cells within the TME

Adenosine receptors and CD73 are also present on cancer-associated fibroblasts (CAFs). One study showed that A2_A_ receptor inhibition in NSCLC impaired CAF and tumor proliferation, which indicates the potential role of CAFs in the TME [67,68].

The adenosine pathway also plays a crucial role in forming blood and lymphatic vessels. Inhibition of CD73 decreased tumor angiogenesis through reduced VEGF secretion. Inhibition of the A2_A_ receptor also resulted in less lymphangiogenesis and consequently reduced lymph node metastases [69,70].

Thus, eADO leads to a TME in favor of tumor growth and invasion.

## 4. Preclinical Data on Adenosine Pathway Inhibition

Several adenosine pathway inhibitors underwent preclinical testing. We will provide an overview of the available preclinical data regarding the most relevant molecules.

### 4.1. CD73 Inhibitors

The first class of molecules are CD73 inhibitors. Here, we review the preclinical evidence of MEDI9447 (oleclumab), CPI-006 (mupadolimab), and PTI199. Oleclumab is a human IgG1λ monoclonal antibody that inhibits the enzymatic function of CD73 in a dual manner [71]. Hay et al. showed that oleclumab restored antitumoral immunity in the TME in a preclinical setting. In a colon carcinoma tumor model, oleclumab inhibited tumor growth and resulted in an increase of CD8^+^ cells in the TME. Combination PD-1 and CD73 inhibition had synergistic effects, leading to a more pronounced tumor reduction [72].

Mupadolimab is also a human monoclonal anti-CD73 antibody that is able to activate human B cells, differentiating it from oleclumab. Mupadolimab also enhances antigen-specific humoral responses, contributing to specific antitumoral immunity [73].

Another CD73 inhibitor, PT199, is a next-generation humanized monoclonal antibody. PT199 has a theoretical advantage over the former molecules due to the ability to completely inhibit CD73 in both its active and inactive enzyme states [74].

### 4.2. CD39 Inhibitors

The next class of agents includes the CD39 inhibitors. TTX-030 is a human monoclonal antibody that blocks CD39. TTX-030 was able to preserve extracellular ATP and decrease the amount of eADO, leading to the proliferation of CD4^+^ and CD8^+^ T cells and an increased secretion of inflammatory cytokines in vitro. Tumor growth reduction was seen in a mouse model [75].

### 4.3. A2_A_ Receptor Antagonists

Several A2_A_ receptor inhibitors are of interest, such as PBF-509 (taminadenant), AZD4635 (imaradenant), CPI-444 (ciforadenant), EOS-850 (inupadenant) and TT-10.

Taminadenant is an oral small molecule A2_A_ receptor inhibitor (A2ARI). Treatment with taminadenant significantly reduced tumor growth in a syngeneic mouse model. The same study showed a restoration of TILs in resected human NSCLC samples upon treatment with taminadenant and PD-L1 inhibition [76].

Imaradenant is another small molecule A2ARI. Preclinical evaluation showed a significant reduction in tumor growth in a mouse model, especially when combined with an immune checkpoint inhibitor targeting PD-L1. This study also demonstrated that imaradenant has the potential to restore cytotoxic T-cell function by reversing the inhibition of IFN-γ signaling and also shows signals of improved T-cell priming by enhancing CD103^+^ DC function [77].

Ciforadenant is another small molecule A2ARI. Ciforadenant treatment resulted in reduced tumor growth in mouse models through activation of CD8^+^ T cells. This activation occurred by downregulating other immune checkpoints, especially LAG-3 and PD-1. Contrary to other preclinical studies, there were no apparent signs of impairment in T cell memory recall [52]. A second preclinical trial showed promising in vivo results by combining ciforadenant with an anti-PD-(L)1 and anti-CTLA-4 antibody. Tumor growth reduction was seen in reduced doses of PD-L1 and CTLA-4 inhibition when treated with this triple combination. This can be of interest because of the toxicity of combined PD-L1 and CTLA-4 blockade. A dose reduction could potentially reduce the risk of side effects [78].

The next molecule is inupadenant, which is also a small molecule A2ARI. Inupadenant differs from the other A2_A_ blockers in its prolonged inhibition of the A2_A_ receptor. Moreover, its inability to penetrate the blood–brain leads to a safer profile toward central neurologic adverse events. Inupadenant was able to reverse adenosine-mediated inhibition of cytokine secretion by T cells in preclinical models [79].

TT-10 is a last A2ARI. Only limited preclinical data are available currently, but these data also showed reduced tumor growth in mouse models. This tumor growth reduction was superior when compared to PD-1 inhibition alone [80].

### 4.4. A2_B_ and Dual A2/A3 Receptor Antagonists

PBF-1129 is a first-in-class orally available selective A2_B_ receptor inhibitor. Preclinical research showed that treatment with PBF-1129 reduced tumor growth, specifically in EGFRm NSCLC. A mouse EGFRm lung cancer model showed that the combination of erlotinib and PBF-1129 led to a delayed recurrence compared to treatment with erlotinib alone [81].

TT-4 and TT-702, both A2_B_ receptor inhibitors, showed preclinical evidence of tumor growth reduction in mouse models [80,82].

AB928 (etrumadenant) is a dual A2_A_/A2_B_ receptor inhibitor. Seifert et al. investigated the effect of etrumadenant on chimeric antigen receptor (CAR) T cell function and found that it potentially improves CAR T cell responses. For instance, the release of granzymes was upregulated in effector T cells, and CAR T cells were also more active when treated with etrumadenant [83]. This suggests that etrumadenant can modify the TME, thereby enhancing antitumor immunity. These findings imply a potential role for combining adenosine inhibition with adoptive cell transfer (ACT) therapies.

M1069 is a second dual A2_A_/A2_B_ receptor inhibitor. In a mouse model, the antitumor immune response to M1069 was more pronounced than that of a selective A2_A_ receptor antagonist [84].

The last class of molecules are the A3 receptor inhibitors. Only one molecule has been tested in the context of cancer to date, more specifically liver cancer. Treatment with CF102 resulted in less liver inflammation and a reduction of liver tumor growth [85].

## 5. Clinical Data on Adenosine Pathway Inhibition

Table 1 contains a list of all adenosine pathway inhibitors that already underwent clinical testing. Some trials are already fully executed, and others are still ongoing. We will focus on the trials that include NSCLC patients.

### 5.1. CD73 Inhibition

#### 5.1.1. Oleclumab/MEDI9447

The recently published phase Ib/II trial by Kim et al. focused on the safety, tolerability, and potential antitumor effects of oleclumab combination therapy [86]. Among others, the combination with osimertinib was evaluated in patients with T790M-negative EGFRm LUAD experiencing disease progression on first- or second-line EGFR TKI. A significant proportion of patients in both dose levels experienced a grade 3 or 4 adverse event (AE) (mainly nail toxicity, neutropenia, and hypertension). There were no dose-limiting toxicities (DLTs) or treatment-related deaths. The efficacy was rather moderate as the ORR was only 11.8% (for the second dose level group), suggesting that combining oleclumab with osimertinib in this patient population had little to no additional benefit. Remarkably, when patients with T790M positivity on circulating tumor DNA but negative T790M analysis on tissue were excluded, a substantial prolongation of progression-free survival (PFS) was noted when compared to historical data (7.4 months vs. 2.8 months). The significance of this finding remains unclear and requires further investigation.

Another trial of interest is the phase II COAST trial. This open-label platform trial investigated the benefit of adding oleclumab or NKG2A inhibition (monalizumab) to the PACIFIC regimen for unresectable stage III NSCLC [87,88]. Patients without disease progression after definitive concomitant chemoradiotherapy (cCRT) were randomized to receive durvalumab alone (n = 67), durvalumab + oleclumab (n = 60), or durvalumab + monalizumab (n = 62). The primary endpoint ORR was numerically higher in the experimental arms with rates of 30% [18.8–43.2] for oleclumab, 35.5% [23.7–48.7] for monalizumab, and 17.9% [23.7–48.7] for durvalumab alone. This result has to be nuanced as ORR is not the optimal endpoint to evaluate the efficacy of adjuvant ICIs post-cCRT, and outcome parameters such as progression-free survival are more suitable to evaluate additional antitumor effects in this setting. Secondary endpoints, which are more reflective of true added clinical activity, also favored the experimental combination. The disease control rate (DCR) at week 16 was significantly higher in both experimental arms with values of 80% [67.7–89.2], 77.4% [65–87.1], and 55.2% [42.6–67.4], respectively. The median PFS (mPFS) was 6.3 months in the control group. mPFS was 15.1 months in the durvalumab + monalizumab arm (HR 0.42 [0.24–0.72]) and was not reached in the durvalumab + oleclumab arm (HR 0.44 [0.26–0.75]). The safety profile was favorable, as none of the experimental arms showed any additional toxicity. It is important to note that ORR and PFS values for the control (durvalumab) arm, when compared to the original PACIFIC trial, were lower than expected. This could be due to different baseline characteristics between those two trials [87,88]. PACIFIC-9, a phase III trial, will prospectively evaluate oleclumab/monalizumab combinations with durvalumab after cCRT in stage III unresectable NSCLC [89].

The recently published phase II NeoCOAST trial is very similar to the previous one and showed very promising results as well [90]. Stage IA3—IIIA resectable NSCLC patients were randomized to receive neoadjuvant durvalumab alone (n = 27), durvalumab + oleclumab (n = 21), durvalumab + monalizumab (n = 20), or durvalumab + danvatirsen (n = 16), an anti-STAT3 antisense oligonucleotide. The major pathological response (MPR) rate was used as the primary outcome, with values of 11.1% [2.4–29.2], 19% [5.4–49.1], 30% [11.9–54.3], and 31.3% [11–58.7], respectively. The medication was considered safe since the number of grade 3 AEs was similar among all four treatment groups.

#### 5.1.2. Mupadolimab/CPI-006

Mupadolimab was tested in a phase I trial in combination with the aforementioned A2ARI ciforadenant in patients with advanced cancers, including NSCLC. A total of 11 patients received mupadolimab monotherapy, and 6 received the combination with ciforadenant. The treatment was considered safe as no adverse events leading to treatment discontinuation were noted. No partial or complete responses were noted in this phase I trial. Specific data regarding the NSCLC patients are not available [91].

#### 5.1.3. BMS-986179

BMS-986179 is currently being researched in advanced malignancies (including NSCLC) in a phase I/IIa trial in combination with nivolumab. In total, 15% of all patients experienced grade 3 AEs, leading to treatment discontinuation in 1 patient. Partial responses (PRs) were noted in 7 patients (11.9%), and 10 patients (16.9%) obtained stable disease (SD). Specific data regarding the NSCLC patients are currently not available [92].

#### 5.1.4. Uliledlimab/TJD5

Uliledlimab was tested as a first-line treatment in combination with toripalimab (anti-PD-1) for non-driver mutated NSCLC in a phase Ib/II trial. A total of 66 patients were enrolled. Grade 3 AEs were noted in 15.2% of patients, with one AE leading to treatment discontinuation. The ORR was 31.3%, and the DCR was 79.2%. Subanalysis showed the highest ORR (57.1%) in patients with a high CD73 score and PD-L1 TPS of 1% or more (n = 14) [93]. A biomarker-guided follow-up trial is currently under preparation but has yet to be initiated.

#### 5.1.5. NZV930

NZV930 was tested in a phase I/Ib trial in patients with advanced cancers (including 8 NSCLC patients) after disease progression on first-line treatment. Patients were treated with NZV930 monotherapy, NZV930 combined with spartalizumab (PD-1 inhibitor) or the A2ARI taminadenant, or the triple combination. NZV930 was considered safe. DLTs were seen in a small subset of patients, and 14% of patients experienced grade 3 toxicity. The clinical trial was suspended because little clinical benefit was observed. The ORR was 0%, and only 11% of the patients experienced SD [94].

### 5.2. A2_A_ Receptor Inhibition

#### 5.2.1. Taminadenant/PBF509/NIR178

Taminadenant has been tested in several phase I trials and is currently being tested in phase II trials. In a phase I/Ib dose escalation and expansion trial focused on NSCLC patients treated with at least one prior line of therapy, patients received either taminadenant alone (n = 25) or in combination with spartalizumab (n = 25) [95]. The primary endpoint was the determination of the maximum tolerated dose. Safety and clinical efficacy were analyzed as secondary endpoints. Grade 3 AEs were reported in 13 patients in total. The treatment was discontinued in 3 patients receiving monotherapy and 5 patients receiving the combination treatment. The ORR was 9.5% in the monotherapy group and 8.3% in the combination arm. A follow-up phase II trial focused on optimization of the dosing regimen, as preclinical data demonstrated a better antitumor effect when taminadenant was intermittently dosed instead of continuously. In total, 62 NSCLC patients received taminadenant continuously or intermittently, in combination with spartalizumab. However, no clinical added value of an intermittent dosing schedule was noted compared to continuous dosing [96].

#### 5.2.2. Imaradenant/AZD4635

The safety and antitumor activity of imaradenant was evaluated in a phase Ia/Ib dose escalation and expansion trial in patients with advanced cancer, including patients with NSCLC experiencing disease progression during or after previous treatment with ICIs [97]. A total of 30 patients with NSCLC were treated in the dose expansion phase of the trial. Patients received either imaradenant as monotherapy (n = 17) or in combination with durvalumab (n = 13). The primary objective of the trial was safety and tolerability. Grade 3 AEs were noted in both arms of NSCLC patients; however, none of them led to treatment discontinuation. A higher amount of grade 3 AEs was noted in the patient group with castration-resistant prostate cancer (n = 108), especially among those treated in the combination arm. Antitumor activity was measured as a secondary endpoint. Unfortunately, no NSCLC patients had an objective response. The DCR was 33.3% in the monotherapy group and 26.7% among the patients treated with the combination treatment. Antitumor activity was more pronounced in prostate cancer patients (ORR: 16.7% in the combination arm and 5% in the monotherapy arm). At this moment, the efficacy of imaradenant is being evaluated in phase II trials focusing on prostate cancer [97].

#### 5.2.3. Ciforadenant/CPI-444

The safety and efficacy of ciforadenant have been researched in several phase I trials. One phase I trial with a specific expansion cohort for NSCLC patients (n = 26) evaluated the safety and clinical efficacy of ciforadenant as monotherapy or in combination with atezolizumab (PD-L1 inhibitor) [98]. Patients had to have experienced disease progression on at least one prior treatment regimen. Ciforadenant was considered safe as no DLTs or treatment discontinuations were noted. The DCR for NSCLC patients was 36% in the monotherapy arm and 71% in the combination arm.

The phase Ib/II MORPHEUS-NSCLC trial compared the combination of atezolizumab and ciforadenant to docetaxel as second-line therapy in NSCLC patients (n = 29) [99]. The primary endpoints were safety and ORR. The experimental treatment was considered safe as there were no adverse events that led to death or treatment discontinuation. Five patients in the control arm experienced adverse events that led to death (n = 1) or treatment discontinuation (n = 4). In terms of clinical efficacy, the ORR was only 6.7% in the experimental arm and 21.4% in the control arm. Further studies focused on prostate cancer and renal cell cancer. Promising results were noted, especially in the latter [100,101].

#### 5.2.4. Inupadenant/EOS850

Inupadenant monotherapy was evaluated in a phase I dose escalation trial. Eligible patients had advanced cancers and disease progression after at least one prior line of therapy [102]. In total, 42 patients were included, including 21 in the dose-finding phase and 21 in the expansion cohort. Safety and tolerability were the primary endpoints. Inupadenant was considered safe. Seven AEs resulted in treatment discontinuation; no dose reductions were noted. Antitumor efficacy was evaluated as a secondary endpoint. The ORR was 4.8%, and the DCR was 33.3%. Currently, a follow-up phase II trial is being conducted in LUAD patients with disease progression on first-line PD-1 inhibiting immunotherapy after having previously experienced clinical benefit and who are chemotherapy-naïve [103]. The trial is made up of 2 stages. The first is a dose-finding stage in which escalating doses of inupadenant are tested in combination with carboplatin and pemetrexed, followed by dose expansion. In stage 2, patients are randomized to receive carboplatin and pemetrexed combined with either inupadenant or placebo. PFS is the primary endpoint of the second stage of the trial.

### 5.3. A2_B_ Receptor Inhibition

#### PBF-1129

PBF-1129 was evaluated in NSCLC patients in a phase I trial after disease progression on standard-of-care therapies [81]. Safety and tolerability were the primary endpoints, and ORR was a secondary endpoint. In total, 21 patients were enrolled. Three patients experienced grade 3 AEs, but none of these led to treatment discontinuation. The ORR was 0%, and 3 patients had SD as a best response. A follow-up trial looking at the combination of PBF-1129 with ICIs is ongoing.

### 5.4. Dual A2_A_ and A2_B_ Receptor Inhibition

#### Etrumadenant/AB928

The ARC-4 trial is a phase I/Ib trial in which etrumadenant, PD-1 inhibition, and chemotherapy (carboplatin-pemetrexed) were evaluated in chemotherapy and ICI treatment-naive patients with NSCLC. The primary endpoint was safety and tolerability. Clinical efficacy was evaluated as a secondary endpoint. A total of 11 patients received etrumadenant combined with platinum doublet and PD-1 inhibitor (pembrolizumab or zimberelimab), including 7 in the dose-finding phase and 4 in the expansion phase. Two patients experienced grade 4 AEs, but no AEs led to treatment discontinuation. Eight patients could be assessed post-baseline, and 4 of them showed a PR [104]. This study is part of 4 phase I trials assessing etrumadenant in different advanced malignancies. The molecule was considered safe in the other studies as well. One DLT was reported [105]. Follow-up phase II trials have produced disappointing results, leading to the withdrawal of the drug from further research [106,107].

**Table 1 cancers-16-03142-t001:** Overview of finished and ongoing trials of drugs targeting the adenosinergic pathway.

Molecule	Target	Phase	Trial Setup	Target Population	NSCLC /Total	Primary Endpoints	Secondary Endpoints	Reference
MEDI9447 Oleclumab	CD73	I	Monotherapy, dose finding	Advanced malignancies, refractory to SOC	0/6	Safety: 1 grade 3 AE, no dose reductions, no deaths	ORR and DCR at week 8: ORR 0%; DCR 0%	Kondo et al. [108]
MEDI9447 Oleclumab	CD73	I	Monotherapy (dose escalation) or oleclumab + durvalumab	Advanced malignancies, at least 1 prior line of therapy; NSCLC had to be EGFRm	42/192 (only oleclumab + durvalumab arm)	Safety and optimal dosing: grade 3/4 AEs in 14.5% of all patients; 1 treatment-related death in the colorectal group; fatigue and rash most common in the NSCLC group	ORR in the NSCLC group: 9.5% (4 PR); 6-month PFS rate in the NSCLC group: 16%	Bendell et al. [109]
MEDI9447 Oleclumab	CD73	II	Durvalumab mono vs. durvalumab + oleclumab vs. durvalumab + monalizumab	Stage III unresectable NSCLC, no progression after cCRT	189/189	ORR: 30% (D + O); 35.5% (D + M); 17.9% (D mono)	DCR at week 16: 81.7% (D + O), 77.4% (D + M), 58.3 (D mono); Median PFS: NR (D + O), 15.1 m (D + M), 6.3 m (D mono); Safety: similar toxicity in all 3 arms, 4 deaths due to study drug	Herbst et al. [87]
MEDI9447 Oleclumab	CD73	II	Neoadjuvant therapy. Durvalumab mono vs. durvalumab + oleclumab vs. durvalumab + monalizumab vs. durvalumab + danvatirsen	Stage IA3—IIIA resectable NSCLC	84/84	MPR: 11.1% (D mono), 19% (D + O), 30% (D + M), 31.3% (D + D)	Safety: no added toxicity of the combination groups compared to monotherapy	Cascone et al. [90]
MEDI9447 Oleclumab	CD73	Ib/II	Oleclumab + osimertinib, dose finding	Advanced EGFRm and tissue T790M-negative NSCLC, progression on a 1/2 gen TKI	26/26	Safety: 1 treatment discontinuation due to pneumonitis, no grade 4/5 TRAEs; ORR: PR in 6 patients, higher ORR and DCR in patients negative for T790M on both tissue and ctDNA	/	Kim et al. [86]
MEDI9447 Oleclumab	CD73	II	NACT + SABR combined with durvalumab or durvalumab + oleclumab	Operable high-risk luminal B breast cancer	0/136	Residual cancer burden on surgical specimen, results ongoing	ORR primary tumor, ORR pathologic lymph nodes; Safety: results ongoing	De Caluwé et al. [110]
CPI-006 Mupadolimab	CD73	I	Dose escalation: mupadolimab monotherapy and in combination with ciforadenant	Relapsed advanced cancers	?/17	Optimal dosing and safety: no DLTs	Tumor reduction seen in 1 patient with prostate cancer; favorable effect on peripheral lymphocytes	Luke et al. [91]
TJD5 Uliledlimab	CD73	Ib/II	2 different doses of uliledlimab in combination with toripalimab	Treatment naïve NSCLC without driver mutations	66/66	Safety: grade 3 AEs in 15.2% of patients; 1 treatment discontinuation	ORR: 31.3% and 50% in CD73 high cohort; DCR: 79.2%	Zhou et al. [93]
NZV930 *	CD73	I	Dose escalation: NZV930 monotherapy vs. combination with spartalizumab vs. combo with taminadenant vs. combo with S + T	Advanced cancers, progression on standard therapy	8/105	Safety: grade 3 AEs in 14% of patients, DLTs in 6.7% of patients	ORR: 0%; DCR: 11%	Fu et al. [94]
BMS-986179	CD73	I/IIa	BMS-986179 + Nivolumab, after 2-week monotherapy lead-in	Advanced cancers	?/59	Safety: grade 3 AEs in 15% of patients; 1 treatment discontinuation; no grade 4/5 AEs reported	ORR: 11.8%; DCR: 28.8%	Siu et al. [92]
TTX-030	CD39		TTX-030 + budigalimab + FOLFOX	Locally advanced or metastatic gastric/gastro-esophageal junction carcinoma	0/44	Safety: grade 3/4 AEs in 11% of patients; no grade 5 toxicity	ORR: 61%	Wainberg et al. [111]
CPI-444 Ciforadenant	A2_A_ receptor	Ib/II	Ciforadenant + atezolizumab vs. docetaxel	Advanced NSCLC, progression on platinum-doublet and PD-(L)1 inhibition	29/29	ORR: 6.7% (C + A), 21.4% (D); Safety: no grade 5 AEs or AEs leading to treatment discontinuation in the experimental arm	Median PFS: 2.3 m (C + A); 3.2 m (D)	Felip et al. [99]
CPI-444 Ciforadenant	A2_A_ receptor	I/Ib	Ciforadenant mono and ciforadenant + atezolizumab	Advanced cancers (including NSCLC), at least 1 and no more than 5 prior therapies	26/34	Safety: 1 grade 3 AE, no dose reductions, no deaths; DCR at week 8: 36% (C mono), 71% (C + A)	/	Fong et al. [98]
CPI-444 Ciforadenant	A2_A_ receptor	I/Ib	Ciforadenant mono and ciforadenant + atezolizumab	Advanced RCC, progression on at 1 least 1 prior therapy	0/68	Safety: 9 grade 3/4 AEs, no dose reductions, no deaths; DCR at month 6: 17% (C mono), 39% (C + A)	/	Fong et al. [100]
CPI-444 Ciforadenant	A2_A_ receptor	I/Ib	Ciforadenant mono and ciforadenant + atezolizumab	Advanced mCRPC, progression on at least 1 prior therapy	0/33	Safety: 2 grade 3/4 AEs, no dose reductions, no deaths; ORR: 0% (C mono), 1 PR in the combination arm	/	Harshman et al. [101]
PBF509/NIR178 Taminadenant	A2_A_ receptor	I/Ib	Taminadenant mono and taminadenant + spartalizumab	Advanced NSCLC, at least 1 prior line of therapy	50/50	Determination of DLTs and MTD	Safety: 13 grade 3/4 AEs among both arms, 3 SAEs among both arms leading to treatment discontinuation; DCR at data cutoff: 42.9% (T mono), 66.7% (T + S)	Chiappori et al. [95]
PBF509/NIR178 Taminadenant	A2_A_ receptor	II	Taminadenant continuous vs. taminadenant intermittent 2 weeks vs. taminadenant intermittent 1 week combined with spartalizumab	Advanced NSCLC, ICI-naïve, 1–3 prior lines of therapy	62/62	ORR: 9% (C); 0% (Int2); 10% (Int1)	Safety: 1 grade 3/4 AE in each treatment arm, no treatment discontinuation or deaths	Lin et al. [96]
AZD4635 Imaradenant	A2_A_ receptor	Ia/Ib	Imaradenant and Imaradenant + durvalumab	Dose expansion phase: 1 cohort NSCLC post-ICI	30/250	Safety: 3 grade 3 AEs among both arms, 1 event of sudden death in colorectal cohort	ORR: 0% in both arms; DCR at 22 weeks: 6.7% (I), 20% (I + D)	Lim et al. [97]
AZD4635 Imaradenant	A2_A_ receptor	I	Imaradenant monotherapy	Advanced malignancies, at least 1 prior line of therapy	0/10	Safety: no grade 3 AEs, no dose reduction, 2 AEs leading to a temporary dose interruption	ORR: 0%; DCR at week 15: 0%	Matsubara et al. [112]
EOS-850 Inupadenant	A2_A_ receptor	I	Inupadenant monotherapy	Advanced malignancies, dose expansion trial	?/42	Optimal dosing; Safety: 7 SAEs leading to treatment discontinuation, no dose reductions	ORR: 4.8%; DCR: 33.3%	Buisseret et al. [102]
EOS-850 Inupadenant	A2_A_ receptor	II	Part 1: carboplatin + pemetrexed + inupadenant dose finding—Part 2: C + P + inupadenant vs. C + P + placebo	NSQ metastatic NSCLC, chemo-naïve and progressive on ICI	40 + 150/190	Part 1: RP2D, results pending; Part 2: PFS, results ongoing	ORR, OS, and AEs: results ongoing	O’Brien et al. [103]
PBF-1129	A2_B_ receptor	I	Dose escalation trial of PBF-1129 monotherapy	Advanced NSCLC, progression on chemotherapy and immunotherapy	21/21	Safety: no DLTs, 3 grade 3 AEs	ORR: 0%—DCR: 14.2%—PFS: 1.5 months—mOS: 4.6 months	Evans et al. [81]
AB928 Etrumadenant	Dual A2_A_ and A2_B_ receptor	I/Ib	Dose finding: etrumadenant + carbo-pem-pembro; dose expansion: etrumadenant + carbo-pem-zimberelimab	Ph 1: NSCLC with genetic alteration and chemo-ICI naïve; Ph Ib: EGFRm	11/11	Safety: 2 SAEs were noted	PR was achieved in 4 patients in total	Spira et al. (ARC-4) [104]
AB928 Etrumadenant	Dual A2_A_ and A2_B_ receptor	II	Zimberelimab vs. domvanalimab + zimberelimab vs. domvanalimab + zimberelimab + etrumadenant	Treatment naïve NSCLC without driver mutations	149/149 (133 patients analyzed)	ORR: 12% (Z), 18% (D + Z), 18% (D + E + Z); mPFS: 5.4 m (Z), 12 m (D + Z), 10.9 m (D + E + Z)	Safety: grade 3 AEs in 58% (Z), 47% (D + Z), and 52% (D + E + Z) of patients	Johnson et al. (ARC-7) [107]
AB928 Etrumadenant	Dual A2_A_ and A2_B_ receptor	II	Domvanalimab + zimberelimab + sacituzumab govitecan vs. domvanalimab + zimberelimab + etrumadenant	Treatment naïve NSCLC without driver mutations	69 to 289 patients to be enrolled	ORR: results ongoing	PFS, OS, and safety: results ongoing	Spira et al. (VELOCITY-lung) [106]
CF-102 Namodenoson	A_3_ receptor	II	Namodenoson vs. placebo	Hepatocellular carcinoma in Child B cirrhosis	0/78	mOS: 4.1 m (N) vs. 4.3 m (P)	ORR: 9% (N) vs. 0% (P); Safety: no treatment discontinuations or deaths	Stemmer et al. [113]

Abbreviations: AE: adverse event; AZ: Astra Zeneca; cCRT: concurrent chemoradiotherapy; ctDNA: circulating tumor DNA; DCR: disease control rate; DLT: dose-limiting toxicity; EGFRm: epidermal growth factor receptor mutated; ICI: immune checkpoint inhibition; mCRPC: metastatic castration-resistant prostate cancer; MPR: major pathological response rate; MTD: maximum tolerated dose; NACT: neo-adjuvant chemotherapy; NSCLC: non-small cell lung cancer; NSQ: non-squamous; ORR: objective response rate; OS: overall survival; PFS: progression-free survival; PR: partial response; RCC: renal cell cancer; RP2D: recommended phase 2 dose; SAE: serious adverse event; SOC: standard of care; TKI: tyrosine kinase inhibitor; TRAE: treatment-related adverse event.

## 6. Combination Strategies

There is a strong rationale for the combination of adenosine targeting agents and other molecules. The forementioned clinical trials show only modest effects when adenosine inhibitors are used as monotherapy. Simultaneous inhibition of multiple steps in the adenosinergic pathway has shown better effects than monotherapy alone in preclinical models [114]. This was also seen in some of the earlier mentioned clinical trials [86,91]. Since there is crosstalk between adenosine and other immune checkpoints, combination strategies targeting different pathways are promising [52]. A good example is the COAST trial showing hopeful results through the combination of CD73 inhibition and PD-1 blockade [87]. There is also in vitro evidence on combining chemotherapy with adenosine inhibiting agents. Gemcitabine and platinum, both frequently used in NSCLC, are known to upregulate CD73 and CD39 in preclinical models [115,116].

Lastly, there are also interactions between thoracic radiotherapy and CD73. In irradiated CD73 knock-out mice, reduced amounts of radiofibrosis have been observed [117].

This shows that there is a scientific background for combination strategies. Since adenosine targeting agents only show limited effects in clinical trials, it seems that combination strategies will be of more clinical importance in the future.

## 7. Conclusions

The adenosinergic pathway plays an important role within the TME. Adenosine has several tumor promoting effects, facilitating tumor growth and metastatic spread, as well as important immunosuppressive effects on multiple immune cells. This results in a less immunogenic TME, leading to reduced ICI efficacy. This is especially clear in driver-mutated LUAD, where the adenosinergic pathway is upregulated, contributing to the development of therapy resistance.

These findings have led to the development of several drugs targeting the adenosinergic pathway, with the A2_A_ receptor and CD73 serving as the most important targets. Several phase I trials of compounds targeting the adenosinergic pathway showed favorable safety profiles.

While monotherapy with adenosine inhibition resulted in limited tumor activity (in particular in the ICI-resistant setting), the true benefit likely lies in combination regimens with other ICIs and chemotherapy. Several ongoing phase II trials will provide more insight into the added clinical benefit of adenosine inhibition.

The phase II COAST study is particularly interesting as improved clinical efficacy was indicated in patients treated with a combination of CD73 and PD-L1 inhibition compared to PD-L1 inhibition alone for unresectable stage III NSCLC after cCRT. The PACIFIC-9 trial will prospectively assess these findings.

Inhibition of the adenosinergic pathway holds theoretical promise, as supported by available preclinical data across several cancer types. However, how to implement adenosinergic inhibition as a therapeutic strategy in NSCLC patients remains unclear and challenging given the overall fairly moderate results of early-phase clinical research so far. Identifying specific patient groups that might benefit from this treatment strategy (e.g., post-chemoradiotherapy) and finding predictive biomarkers will likely advance the future of therapeutic adenosinergic inhibition.

## Figures and Tables

**Figure 1 cancers-16-03142-f001:**
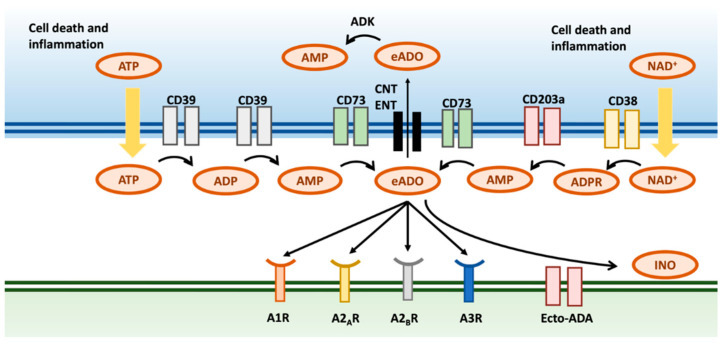
Overview of the canonical and non-canonical pathways, the four primary adenosine receptors, and the main clearance routes of adenosine. Abbreviations: ADP: adenosine diphosphate; ADPR: adenosine diphosphate ribose; AMP: adenosine monophosphate; ATP: adenosine triphosphate, ADK: adenylate kinase; CNT: concentrative nucleoside transporter; Ecto-ADA: adenosine-deaminase; ENT: equilibrative nucleoside transporter; INO: inosine; NAD: nicotinamide adenine dinucleotide.

**Figure 2 cancers-16-03142-f002:**
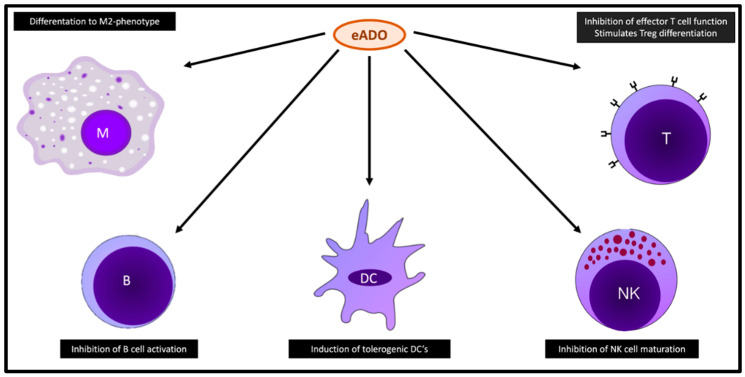
Effects of adenosine on various immune cells. Abbreviations: B: B cell; DC: dendritic cell; M: macrophage; NK: natural killer cell; T: T cell; Treg: regulatory T cell.

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
