# Peer review of "The Adenosinergic Pathway in Non-Small Cell Lung Cancer"

_cancers, 2024, doi:10.3390/cancers16183142_

Round 1

Reviewer 1 Report

Comments and Suggestions for Authors

In this review, the authors present the biological processes and functions of adenosine in vivo in a logical order. The significance of targeting adenosinic pathway in the treatment of non-small cell lung cancer is pointed out, and the current status of clinical research is listed. The content of this review is well organized, informative, and fluent in language. However, the rechecking rate of articles is too high. Therefore, I recommend that the article be accepted after major revisions.

Suggestions for improvement:

-The simple summary and key word sections of the article are missing and the authors need to supplement them when revising.

-iThenticate reports a 23% percent match in the manuscript, necessitating a reduction.

- This article only gives a brief overview of the limitations and development direction (such as combination therapy, etc.) of adenosine pathway in the treatment of non-small cell lung cancer at present, and it is suggested to improve when it is revised.

- Reference section (lines510-511) citation format needs to be confirmed.

Reviewer 2 Report

Comments and Suggestions for Authors

This paper discusses the involvement of adenosinergic pathway in lung cancer pathogenesis and the clinical perspectives of relevant targeted drugs for the treatment of this disease.

It is not immediately clear why the review focuses only on lung cancer, because this pathway is apparently involved in other cancer types.

Many statements are not clearly articulated. For example, lines 127-129: “…CD73, as a promotor of immune suppression, has been studied in oncogene-driven NSCLC, especially in EGFR mutated (EGFRm) LUAD. CD73 also seems to be upregulated in KRAS mutated and ALK re-arranged LUAD, but fewer studies are available on this topic [21]. …” The first sentence apparently means that CD73 is upregulated in EGFR-driven NSCLC, but this is not immediately clear from this wording.

Many sections do not deliver a message, for example 2.4, 2.5, 3.2, 3.3

The text in the section 5 is rather a repetition of the Table 1.

There are multiple inaccuracies; for example, see the list of authors, simple summary, title of the section 2.4, etc.

Comments on the Quality of English Language

The text is perfectly understandable, but still requires thorough editing

Round 2

Reviewer 2 Report

Comments and Suggestions for Authors

The suggestions of the Reviewers have been properly addressed 

Comments on the Quality of English Language

Minor editing is required